# Variable Magellanic HMXB Sources versus Variable ULX Sources: Nothing to Brag about the ULX Sources

**Dimitris M. Christodoulou** [1,*,†] 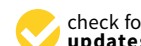, **Silas G. T. Laycock** [1,2,†], **Rigel Cappallo** [1,2,†],
**Ankur Roy** [1,2,†], **Sayantan Bhattacharya** [1,2,†] **and Demosthenes Kazanas** [3,†]

[1] Lowell Center for Space Science and Technology, University of Massachusetts Lowell,
Lowell, MA 01854, USA; silas_laycock@uml.edu (S.G.T.L.); rigelcappallo@gmail.com (R.C.);
ankur2392roy@gmail.com (A.R.); sayantan34@gmail.com (S.B.)

[2] Deptartment of Physics and Applied Physics, University of Massachusetts Lowell, Lowell, MA 01854, USA

[3] NASA Goddard Space Flight Center, Laboratory for High-Energy Astrophysics, Greenbelt, MD 20771, USA;
demos.kazanas@nasa.gov

\* Correspondence: dimitris_christodoulou@uml.edu

† These authors contributed equally to this work.

**Abstract:** We carry out a meta-analysis of ultraluminous X-ray (ULX) sources that show large variabilities (by factors of $> 10$) between their highest and lowest emission states in the X-ray energy range of 0.3–10 keV. We are guided by a recent stringent compilation of 25 such X-ray sources by Song et al. We examine the relation of $\log N$ versus $\log S_{\max}$, where $N$ is the number of sources radiating above the maximum-flux level $S_{\max}$. We find a strong deviation from all previously determined slopes in various high-mass X-ray binary (HMXB) samples. In fact, the ULX data clearly show a slope of $-0.91$. Thus, ULX sources do not appear to be uniform and isotropic in our Universe. We compare the ULX results against the local X-ray luminosity function of HMXBs in the Small Magellanic Cloud (SMC) constructed from our latest library that includes 41 *Chandra* 0.3–8 keV sources and 56 *XMM-Newton* 0.2–12 keV sources. The ULX data are not drawn from the same continuous distribution as the SMC data (the ULX data peak at the low tails of the SMC distributions), and none of our data sets is drawn from a normal distribution or from a log-normal distribution (they all show marked excesses at both tails). At a significance level of $\alpha = 0.05$ ($2\sigma$), the two-sample $p$-value of the Kolmogorov–Smirnov (KS) test gives $p = 4.7 \times 10^{-3} < \alpha$ for the ULX versus the small *Chandra* sample and $p = 1.1 \times 10^{-5} << \alpha$ for the ULX versus the larger *XMM-Newton* sample, respectively. This adds to the evidence that ULX sources are not simply the higher end of the known local Be/X-ray pulsar distribution, but they represent a class of X-ray sources different from the young sources found in the SMC and in individual starburst galaxies. On the other hand, our two main SMC data sets are found to be statistically consistent, as they are drawn from the same continuous parent distribution (null hypothesis $H_0$): at the $\alpha = 0.05$ significance level, the two-sample KS test shows an asymptotic $p$-value of $0.308 > \alpha$, which tells us to accept $H_0$.

**Keywords:** high-mass X-ray binary (HMXB); neutron star; pulsar; ultraluminous X-ray (ULX) source; X-rays

## 1. Introduction

We revisit a new data set of strongly variable ultraluminous X-ray (ULX) sources that was produced by Song et al. [1], and we compare these data (Table 1) statistically to the distribution of Be/X-ray sources produced by our latest library (version 2.0; see Reference [2] for version 1.0) for the Small Magellanic Cloud (SMC) (Tables 1 and 2 below). The SMC sources are all confirmed pulsars

with massive companions, whereas the ULX sources are believed to also host pulsars, and those with confirmed pulsars have all turned out to be high-mass X-ray binaries (HMXBs) [3–7].

Song et al. [1] distilled a set of 25 ULX sources at 0.3–10 keV energies that show strong variabilities by factors of more than 10, similar to virtually all SMC X-ray sources with the notable exception of SMC X-1 [2]. Their selection criteria were extremely stringent, which makes this sample quite valuable for follow-up studies (see Table 1 in Reference [1]). We expect monitoring campaigns to be launched in the coming years to observe members of this exotic group of ULX sources (see also previous fundamental results in References [3–23]). The exotic nature of these sources is evident from their extreme X-ray luminosities as contrasted by their modest X-ray fluxes. This marked contrast seen in the data of Reference [1] motivated us to pursue the present statistical analysis.

There have been numerous attempts to map out the $\log N$-$\log S$ relation in extragalactic X-ray sources including HMXBs, low-mass X-ray binaries (LMXBs), and background active galactic nuclei (AGN) [24–30], to name a few. Here $N$ is the number of sources detected above each particular flux level $S$. In a homogeneous and isotropic universe, the distribution of sources is expected to be uniform, and the flux is expected to drop as $1/d^2$ with radial distance $d$ at a given luminosity, whereas the number $N$ is expected to increase as $d^3$ owing to the uniform density. Then a plot of the $\log N$-$\log S$ relation should exhibit the Euclidean slope of $-3/2$ [24]. This slope is not observed in recent compilations of extragalactic X-ray sources [25,26,29–31] in which contamination from LMXBs and background AGN pulls the observed slopes to much shallower values; for example, typical HMXB samples show slopes in the $-(0.4$–$0.6)$ range [25,31–33].

**Table 1.** SMC SXP fluxes (0.3–8 keV) observed by *Chandra* (ACIS-I)[a].

| Source No. | SXP Name | $S_{max}/10^{-13}$ (erg s$^{-1}$ cm$^{-2}$) | $\|Error\|/10^{-13}$ (erg s$^{-1}$ cm$^{-2}$) | $N_{Obs}$[b] | $S_{min}/10^{-13}$ (erg s$^{-1}$ cm$^{-2}$) | $\|Error\|/10^{-13}$ (erg s$^{-1}$ cm$^{-2}$) | $L_{X,max}/10^{36}$ (erg s$^{-1}$) | $\|Error\|/10^{36}$ (erg s$^{-1}$) |
|---|---|---|---|---|---|---|---|---|
| 1 | 3.34 | 9.04 | 0.42 | 84 | 0.25 | 0.07 | 4.16 | 0.19 |
| 2 | 6.88 | 1.88 | 0.22 | 3 | 1.42 | 0.17 | 0.86 | 0.10 |
| 3 | 7.78 | 53.29 | 1.05 | 5 | 0.52 | 0.15 | 24.51 | 0.48 |
| 4 | 7.92 | 56.69 | 1.05 | 2 | 7.84 | 0.4 | 26.08 | 0.48 |
| 5 | 8.8 | 2.78 | 0.31 | 3 | 2.1 | 0.21 | 1.28 | 0.14 |
| 6 | 9.13 | 10.04 | 0.45 | 7 | 3.75 | 0.27 | 4.62 | 0.21 |
| 7 | 15.3 | 3.54 | 0.29 | 3 | 1.6 | 0.19 | 1.63 | 0.13 |
| 8 | 18.3 | 86.18 | 1.29 | 2 | 3.73 | 0.27 | 39.65 | 0.59 |
| 9 | 22.1 | 0.42 | 0.11 | 1 | $\cdots$ | $\cdots$ | 0.19 | 0.05 |
| 10 | 25.5 | 118.35 | 1.51 | 2 | 1.95 | 0.22 | 54.44 | 0.69 |
| 11 | 46.6 | 0.76 | 0.15 | 3 | 0.13 | 0.05 | 0.35 | 0.07 |
| 12 | 59.0 | 4.84 | 0.34 | 4 | 0.79 | 0.13 | 2.23 | 0.16 |
| 13 | 65.8 | 38.41 | 0.86 | 1 | $\cdots$ | $\cdots$ | 17.67 | 0.40 |
| 14 | 82.4 | 89.75 | 1.31 | 2 | 0.36 | 0.1 | 41.29 | 0.60 |
| 15 | 101 | 8.32 | 0.4 | 1 | $\cdots$ | $\cdots$ | 3.83 | 0.18 |
| 16 | 138 | 14.76 | 0.54 | 3 | 2.97 | 0.24 | 6.79 | 0.25 |
| 17 | 140 | 0.68 | 0.13 | 2 | 0.18 | 0.07 | 0.31 | 0.06 |
| 18 | 152 | 96.98 | 1.38 | 1 | $\cdots$ | $\cdots$ | 44.61 | 0.63 |
| 19 | 153 | 7.97 | 0.4 | 19 | 0.18 | 0.06 | 3.67 | 0.18 |
| 20 | 172 | 28.07 | 0.74 | 3 | 10.88 | 0.46 | 12.91 | 0.34 |
| 21 | 175 | 10.31 | 0.42 | 4 | 4.34 | 0.34 | 4.74 | 0.19 |
| 22 | 214 | 30.11 | 0.78 | 1 | $\cdots$ | $\cdots$ | 13.85 | 0.36 |
| 23 | 264 | 14.79 | 0.53 | 3 | 2.55 | 0.23 | 6.8 | 0.24 |
| 24 | 280 | 7.21 | 0.37 | 2 | 3.59 | 0.31 | 3.32 | 0.17 |
| 25 | 292 | 0.66 | 0.15 | 3 | 0.34 | 0.09 | 0.3 | 0.07 |
| 26 | 304 | 18.9 | 0.61 | 24 | 0.4 | 0.1 | 8.69 | 0.28 |
| 27 | 323 | 20.21 | 0.63 | 4 | 7.3 | 0.38 | 9.3 | 0.29 |
| 28 | 327 | 101.11 | 1.42 | 2 | 82.54 | 1.28 | 46.51 | 0.65 |
| 29 | 342 | 0.27 | 0.1 | 1 | $\cdots$ | $\cdots$ | 0.12 | 0.04 |
| 30 | 348 | 12.04 | 0.49 | 74 | 0.19 | 0.07 | 5.54 | 0.23 |
| 31 | 455 | 18.49 | 0.61 | 18 | 1.98 | 0.19 | 8.51 | 0.28 |
| 32 | 504 | 41.26 | 0.91 | 3 | 9.39 | 0.43 | 18.98 | 0.42 |
| 33 | 523 | 0.51 | 0.12 | 16 | 0.11 | 0.05 | 0.23 | 0.05 |
| 34 | 565 | 52.47 | 1.11 | 4 | 2.29 | 0.21 | 24.14 | 0.51 |
| 35 | 645 | 4.48 | 0.32 | 2 | 2.82 | 0.26 | 2.06 | 0.15 |
| 36 | 701 | 18.14 | 0.59 | 2 | 10.03 | 0.45 | 8.34 | 0.27 |
| 37 | 726 | 8.86 | 0.41 | 53 | 0.11 | 0.05 | 4.08 | 0.19 |
| 38 | 893 | 34.28 | 0.83 | 4 | 0.36 | 0.09 | 15.77 | 0.38 |
| 39 | 967 | 7.38 | 0.38 | 14 | 0.3 | 0.07 | 3.4 | 0.18 |
| 40 | 1062 | 94.25 | 1.34 | 15 | 10.18 | 0.44 | 43.36 | 0.62 |
| 41 | 1323 | 38.05 | 0.87 | 185 | 0.3 | 0.09 | 17.5 | 0.40 |

[a] *Chandra* X-ray fluxes were computed by the procedures described in References [2,15]. X-ray luminosities were determined from Equation (6) below. [b] Number of observations in the *Chandra* archive up to 31-12-2019.

**Table 2.** SMC SXP fluxes (0.2–12 keV) observed by *XMM-Newton* (EPIC PN + MOS1 + MOS2) [a].

| Source No. | SXP Name | $S_{max}/10^{-13}$ (erg s$^{-1}$ cm$^{-2}$) | $\lvert Error \rvert/10^{-13}$ (erg s$^{-1}$ cm$^{-2}$) | $N_{Obs}$ [b] | $S_{min}/10^{-13}$ (erg s$^{-1}$ cm$^{-2}$) | $\lvert Error \rvert/10^{-13}$ (erg s$^{-1}$ cm$^{-2}$) | $L_{X,max}/10^{36}$ (erg s$^{-1}$) | $\lvert Error \rvert/10^{36}$ (erg s$^{-1}$) |
|---|---|---|---|---|---|---|---|---|
| 1 | 0.72 | 8130.74 | 10.27 | 6 | 276.34 | 0.97 | 374.038 | 0.473 |
| 2 | 2.37 | 2913.34 | 3.6 | 4 | 0.05 | 0.02 | 134.022 | 0.166 |
| 3 | 2.76 | 1634.09 | 3.8 | 1 | $\cdots$ | $\cdots$ | 75.173 | 0.175 |
| 4 | 3.34 | 3.31 | 0.48 | 49 | 0.33 | 0.12 | 0.152 | 0.022 |
| 5 | 4.78 | 2162.17 | 4.18 | 1 | $\cdots$ | $\cdots$ | 99.466 | 0.192 |
| 6 | 5.05 | 641.94 | 1.91 | 3 | 77.14 | 0.98 | 29.531 | 0.088 |
| 7 | 6.85 | 548.69 | 3.85 | 1 | $\cdots$ | $\cdots$ | 25.242 | 0.177 |
| 8 | 7.78 | 1482.34 | 5.3 | 5 | 5.3 | 1.21 | 68.192 | 0.244 |
| 9 | 7.92 | 47.33 | 1.05 | 2 | 39.1 | 0.95 | 2.177 | 0.048 |
| 10 | 8.80 | 0.06 | 0.06 | 3 | 0.05 | 0.06 | 0.003 | 0.003 |
| 11 | 8.02 | 5.07 | 0.57 | 14 | 4.31 | 0.12 | 0.233 | 0.026 |
| 12 | 9.13 | 6.3 | 0.46 | 4 | 3.1 | 0.15 | 0.29 | 0.021 |
| 13 | 11.5 | 0.04 | 0.03 | 1 | $\cdots$ | $\cdots$ | 0.002 | 0.001 |
| 14 | 11.9 | 114.38 | 1.2 | 1 | $\cdots$ | $\cdots$ | 5.262 | 0.055 |
| 15 | 15.3 | 1.55 | 0.18 | 2 | 1.39 | 0.37 | 0.071 | 0.008 |
| 16 | 18.3 | 137.49 | 1.93 | 5 | 22.72 | 0.93 | 6.325 | 0.089 |
| 17 | 22.1 | 0.04 | 0.03 | 1 | $\cdots$ | $\cdots$ | 0.002 | 0.001 |
| 18 | 25.5 | 3.75 | 0.22 | 3 | 0.28 | 0.09 | 0.173 | 0.01 |
| 19 | 31.0 | 0.42 | 0.08 | 2 | 0.18 | 0.12 | 0.019 | 0.004 |
| 20 | 46.6 | 96.84 | 1.95 | 5 | 0.05 | 0.02 | 4.455 | 0.09 |
| 21 | 59.0 | 642.07 | 2.28 | 9 | 0.24 | 0.13 | 29.537 | 0.105 |
| 22 | 65.8 | 20.27 | 0.49 | 2 | 10.74 | 0.39 | 0.932 | 0.022 |
| 23 | 74.7 | 19.11 | 0.58 | 5 | 1.51 | 0.24 | 0.879 | 0.027 |
| 24 | 91.1 | 97.91 | 3.46 | 3 | 12.81 | 1.37 | 4.504 | 0.159 |
| 25 | 101 | 8.37 | 0.25 | 1 | $\cdots$ | $\cdots$ | 0.385 | 0.011 |
| 26 | 138 | 12.73 | 0.83 | 7 | 5.86 | 0.17 | 0.586 | 0.038 |
| 27 | 140 | 0.38 | 0.07 | 4 | 0.17 | 0.09 | 0.017 | 0.003 |
| 28 | 152 | 57.07 | 0.89 | 13 | 0.14 | 0.08 | 2.625 | 0.041 |
| 29 | 153 | 14.52 | 0.51 | 1 | $\cdots$ | $\cdots$ | 0.668 | 0.023 |
| 30 | 169 | 24.98 | 0.53 | 2 | 23.54 | 0.65 | 1.149 | 0.025 |
| 31 | 172 | 33.64 | 0.99 | 3 | 10.99 | 1.32 | 1.547 | 0.046 |
| 32 | 175 | 87.91 | 1.24 | 6 | 11.01 | 0.39 | 4.044 | 0.057 |
| 33 | 202A | 89.63 | 2.31 | 20 | 8.46 | 0.71 | 4.123 | 0.106 |
| 34 | 202B | 119.74 | 1.43 | 3 | 6.26 | 0.53 | 5.508 | 0.066 |
| 35 | 214 | 21.17 | 0.7 | 1 | $\cdots$ | $\cdots$ | 0.974 | 0.032 |
| 36 | 264 | 52.24 | 0.89 | 4 | 0.95 | 0.19 | 2.403 | 0.041 |
| 37 | 280 | 68.63 | 1.35 | 15 | 0.51 | 0.15 | 3.157 | 0.062 |
| 38 | 292 | 0.22 | 0.09 | 3 | 0.1 | 0.05 | 0.01 | 0.004 |
| 39 | 293 | 74.85 | 1.21 | 6 | 8.86 | 1.46 | 3.443 | 0.056 |
| 40 | 304 | 22.67 | 1.55 | 26 | 0.5 | 0.11 | 1.043 | 0.071 |
| 41 | 323 | 13.32 | 0.46 | 3 | 8.14 | 0.32 | 0.613 | 0.021 |
| 42 | 327 | 9.83 | 0.8 | 5 | 4.57 | 0.7 | 0.452 | 0.037 |
| 43 | 342 | 2.94 | 0.15 | 3 | 0.4 | 0.13 | 0.135 | 0.007 |
| 44 | 348 | 15.38 | 0.71 | 23 | 1.01 | 1.14 | 0.707 | 0.033 |
| 45 | 455 | 24.08 | 2.05 | 18 | 3.88 | 0.35 | 1.108 | 0.094 |
| 46 | 504 | 16.04 | 0.34 | 2 | 3.71 | 0.38 | 0.738 | 0.016 |
| 47 | 523 | 1.13 | 0.28 | 9 | 0.08 | 0.05 | 0.052 | 0.013 |
| 48 | 565 | 30.28 | 0.79 | 18 | 0.66 | 0.13 | 1.393 | 0.036 |
| 49 | 645 | 14.3 | 0.51 | 7 | 0.42 | 0.11 | 0.658 | 0.023 |
| 50 | 701 | 10.43 | 0.29 | 4 | 0.88 | 0.21 | 0.48 | 0.014 |
| 51 | 726 | 12.83 | 1.53 | 34 | 0.28 | 0.16 | 0.59 | 0.071 |
| 52 | 756 | 20.82 | 0.58 | 3 | 0.7 | 0.35 | 0.958 | 0.027 |
| 53 | 893 | 5.67 | 0.53 | 3 | 0.89 | 0.1 | 0.261 | 0.024 |
| 54 | 967 | 30.01 | 5.5 | 5 | 1.34 | 0.75 | 1.38 | 0.253 |
| 55 | 1062 | 75.42 | 0.5 | 6 | 12.78 | 0.14 | 3.469 | 0.023 |
| 56 | 1323 | 46.9 | 0.69 | 50 | 0.91 | 0.08 | 2.158 | 0.032 |

[a] *XMM-Newton* X-ray fluxes were computed by the procedures described in References. [2,15]. X-ray luminosities were determined from Equation (6) below. [b] Number of observations in the *XMM-Newton* archive up to 18-11-2019.

The faintest X-ray point sources observed are often located in nearby galaxies, a trend that is not followed by all ULX sources [1,34]. This implies that there are effects in the ULX emission and/or systematics in the ULX observations that make them deviate from this distance-dependent expectation. For example, it has been argued by some groups that ULX sources appear to be so powerful because they are beaming in the direction of the observer [22,23,35–37], clearly a selection effect. If this is the case, their log *N*-log *S* diagram should show a strong deviation from the theoretical Euclidean line with slope −3/2 and from the SMC log *N*-log *S* best-fit lines whose slopes fall consistently in the range of −(0.37–0.6) [25], where −0.37 is the slope of the securely identified SMC HMXBs and −0.6 is the mean slope of the examined "AGN-contaminated" samples.

In the next sections, we document the ULX and SMC samples and the log *N*-log $S_{max}$ behaviors of strongly variable ULX sources and the known SMC HMXB sources, where $S_{max}$ is the maximum flux observed among all recorded outbursts of these sources. We choose to focus mostly on X-ray fluxes (rather than on X-ray luminosities) in order to avoid a dependence of results on distances which are

uncertain for ULX sources and their host galaxies; a timid analysis of X-ray luminosities is presented in Section 5. The $-3/2$ theoretical value for the Euclidean universal slope is not borne out of the current data sets, making the case for beaming of ULX sources perhaps even stronger. The measured slopes in our SMC data sets are too shallow (in the range of $-0.39$ to $-0.63$; see Table 3 below), which implies that, in the case of ULX sources with steeper slopes ($-0.91$), we are observing a lot fewer ULX sources at larger fluxes, whereas we see an excess of Magellanic/local-group HMXB sources at larger fluxes. The observed moderate to small ULX fluxes suggest that ULX outbursts are evidently nothing to brag about, as compared to the most powerful (type II) outbursts of SMC HMXB sources [38–40]. In the final two sections, we discuss and summarize our results.

**Table 3.** Least-squares fits to X-ray flux/luminosity functions.

| No. | Data Set | Figure | Slope | $1\sigma$ Error | $y$-Intercept | $1\sigma$ Error | $r$ | $p$ |
|-----|----------|--------|-------|-----------------|---------------|-----------------|-----|-----|
| 1 | ULX | 3 | $-0.9103$ | 0.0574 | 1.5921 | 0.0545 | $-0.9807$ | 0.0401 |
| 2 | SMC Chandra IR [a] | 4 | $-0.6338$ | 0.0281 | 2.0181 | 0.0356 | $-0.9922$ | 0.0282 |
| 2 | SMC Chandra HE [b] | 4 | $-6.9025$ | 0.1996 | 14.0961 | 0.3258 | $-0.9996$ | 0.0184 |
| 3 | SMC XMM-Newton IR [a] | 5 | $-0.4469$ | 0.0139 | 2.0581 | 0.0185 | $-0.9966$ | 0.0198 |
| 3 | SMC XMM-Newton HE [b] | 5 | $-1.2759$ | 0.0631 | 3.6030 | 0.1085 | $-0.9976$ | 0.0315 |
| 4 | SMC Combined IR [a] | $\cdots$ | $-0.3939$ | 0.0195 | 2.0615 | 0.0259 | $-0.9915$ | 0.0315 |
| 4 | SMC Combined HE [b] | $\cdots$ | $-2.0196$ | 0.1331 | 5.1571 | 0.2289 | $-0.9957$ | 0.0419 |
| 5 | All Combined [c] | 11 | $-0.5716$ | 0.0240 | 2.0042 | 0.0529 | $-0.9896$ | 0.0267 |

[a] IR: Intermediate Range of Fluxes. [b] HE: High End of Fluxes. [c] SMC $L_{X,max}$ data and ULX data combined.
Note: $r$ is the correlation and $p$ is the $p$-value of the statistic.

## 2. X-ray Data Sets

We analyze five data sets comparing and combining the Song et al. ULX sources and our library's SMC HMXB sources:

1. The strongly variable ULX data set (25 sources) in the 0.3–10 keV band [1] ;
2. Our library's sample of SMC *Chandra* data (41 sources) in the 0.3–8 keV band (Table 1);
3. Our library's sample of SMC *XMM-Newton* data (56 sources) in the 0.2–12 keV band (Table 2);
4. The combined (2 + 3) SMC data set (58 sources) considering the maximum flux $S_{max}$ for each source;
5. A pseudo-data set (1 + 4) that naively combines the SMC with the ULX luminosities (83 sources).

### 2.1. Histograms

The SMC *XMM-Newton* data set is the largest of the three main sets (1–3). In Figures 1 and 2, we compare it against the other two main samples. The ULX distribution is clearly dissimilar to the SMC *XMM-Newton* distribution. The ULX sample peaks at a much lower flux. The histograms appear to be roughly mirror images of one another due to strong secondary peaks on opposite tails. On the other hand, the *Chandra* and *XMM-Newton* SMC samples appear to be quite similar in shape in Figure 2. Both of them show secondary peaks in the tails, and the strong secondary peaks nearly overlap at the high end (around a logarithmic value of 2).

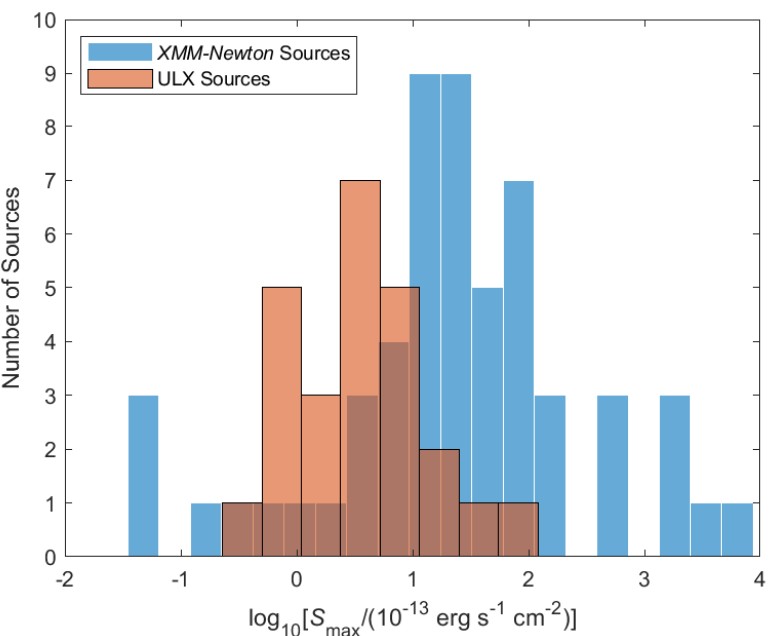

**Figure 1.** Flux histograms. The SMC *XMM-Newton* and ULX data sets appear to be dissimilar. The ULX sample clearly peaks at lower flux values, and its secondary peak is located at the lower tail of the distribution, contrary to the secondary peak of the SMC *XMM-Newton* data set.

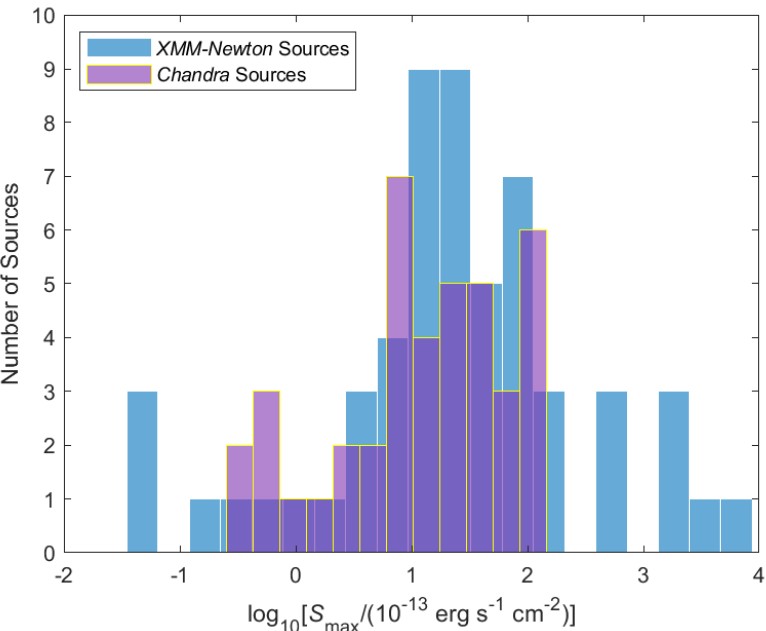

**Figure 2.** Flux histograms. The SMC *XMM-Newton* and SMC *Chandra* data sets appear to be quite similar. In particular, both histograms show secondary peaks in the tails of the distributions.

Because of the presence of secondary peaks in the tails, the data do not appear to be normally distributed in any of these cases. A formal one-sample Kolmogorov–Smirnov (KS) test confirms that none of our samples is drawn from a normal distribution. The results of our statistical calculations (hypothesis testing) are listed in Table 4 below. Another set of tests on the log-values of the maximum fluxes shows that no sample is derived from a log-normal distribution either.

**Table 4.** Kolmogorov–Smirnov Tests.

| Data Sets | $p$ | $D$ | $D_{\text{crit}}$ | Result |
|---|---|---|---|---|
| $H_0$ ($\alpha = 0.05$): Two samples from the same continuous distribution | | | | |
| Chandra, XMM-Newton | $0.308 > \alpha$ | 0.193 | 0.279 | $H_0$ |
| Chandra, ULX | $4.7 \times 10^{-3}$ | 0.425 | 0.345 | $H_1$ |
| XMM-Newton, ULX | $1.1 \times 10^{-5}$ | 0.572 | 0.327 | $H_1$ |
| SMC Combined, ULX | $1.4 \times 10^{-6}$ | 0.616 | 0.325 | $H_1$ |
| $H_0$ ($\alpha = 0.05$): Samples from a normal distribution | | | | |
| ULX | $8.2 \times 10^{-12}$ | 0.703 | 0.264 | $H_1$ |
| Chandra | $8.4 \times 10^{-26}$ | 0.827 | 0.208 | $H_1$ |
| XMM-Newton | $2.9 \times 10^{-37}$ | 0.856 | 0.178 | $H_1$ |
| SMC Combined | $1.9 \times 10^{-40}$ | 0.877 | 0.175 | $H_1$ |
| All Combined [a] | $1.9 \times 10^{-13}$ | 0.500 | 0.175 | $H_1$ |

[a] SMC combined $L_{X,\text{max}}$ data and ULX $L_{X,\text{max}}$ data. Note: $p$ is the $p$-value, $D$ is the KS statistic, and $D_{\text{crit}}$ is its critical value from Equation (4).

### 2.2. Maximum Fluxes and Maximum X-ray Luminosities

Song et al. [1] presented 25 highly variable ULXs in their final Table 1. Flux variability between high and low emission states is more than a factor of 10. The maximum X-ray fluxes $S_{\text{max}}$ and their error bars, which are of interest in this work, are shown in Figure 3. It is evident that the flux errors are small in all of these ULX sources. On the other hand, our library of SMC HMXBs contains 41 *Chandra* sources and 56 *XMM-Newton* sources at comparable energy ranges. Their maximum fluxes and X-ray luminosities are listed in Tables 1 and 2, respectively. The X-ray fluxes are also illustrated in Figures 4 and 5, respectively.

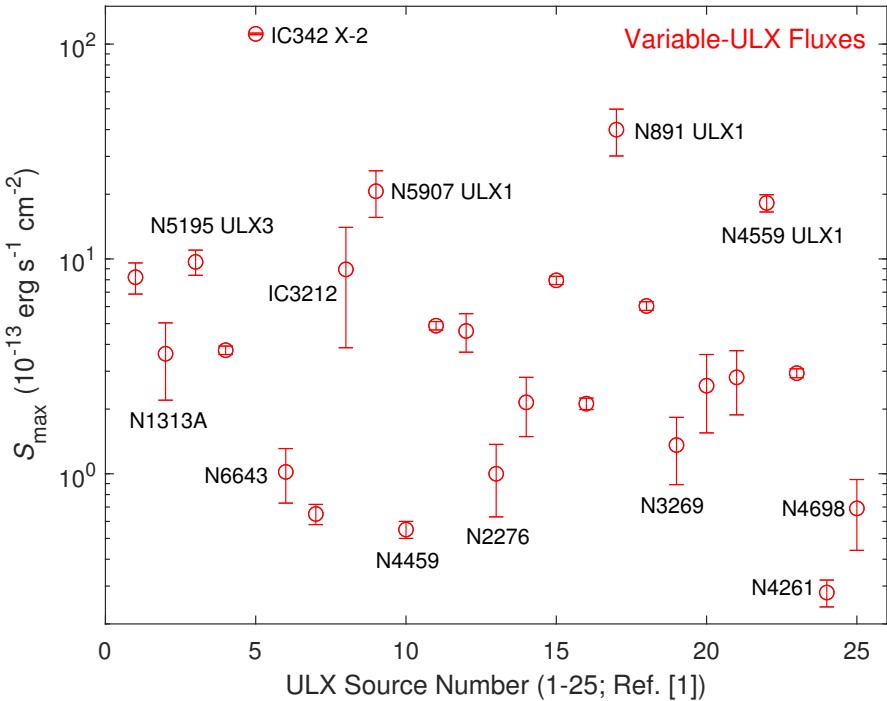

**Figure 3.** Sample of maximum X-ray fluxes for the ULX sources listed in Table 1 of Reference [1]. Errors are small for virtually all sources.

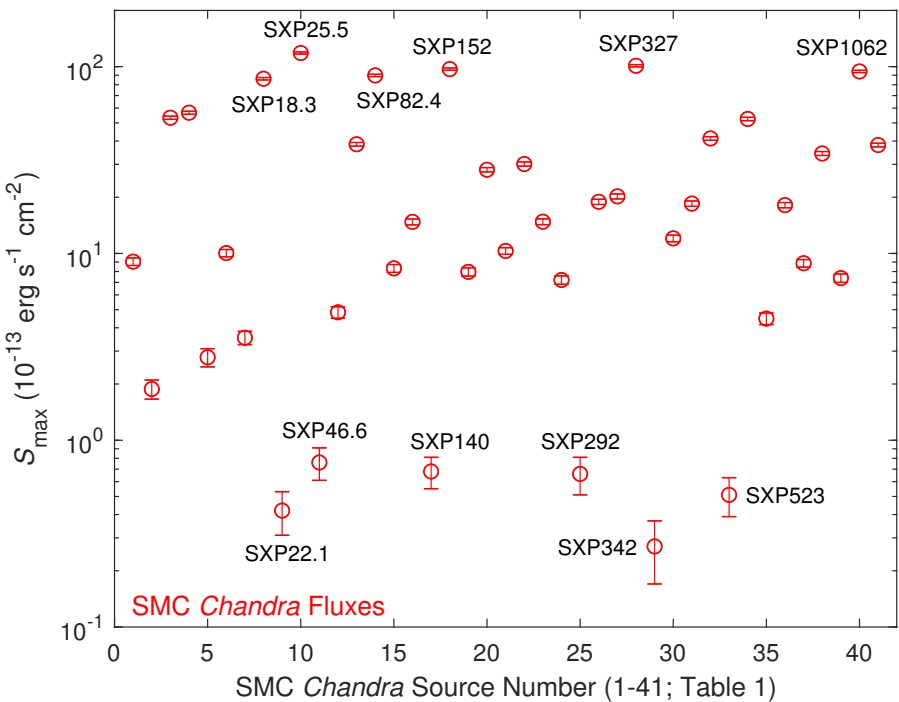

**Figure 4.** Sample of maximum X-ray fluxes for the SMC *Chandra* ACIS-I sample in Table 1. Owing to *Chandra*'s unprecedented accuracy and sensitivity, errors are extremely small for all sources.

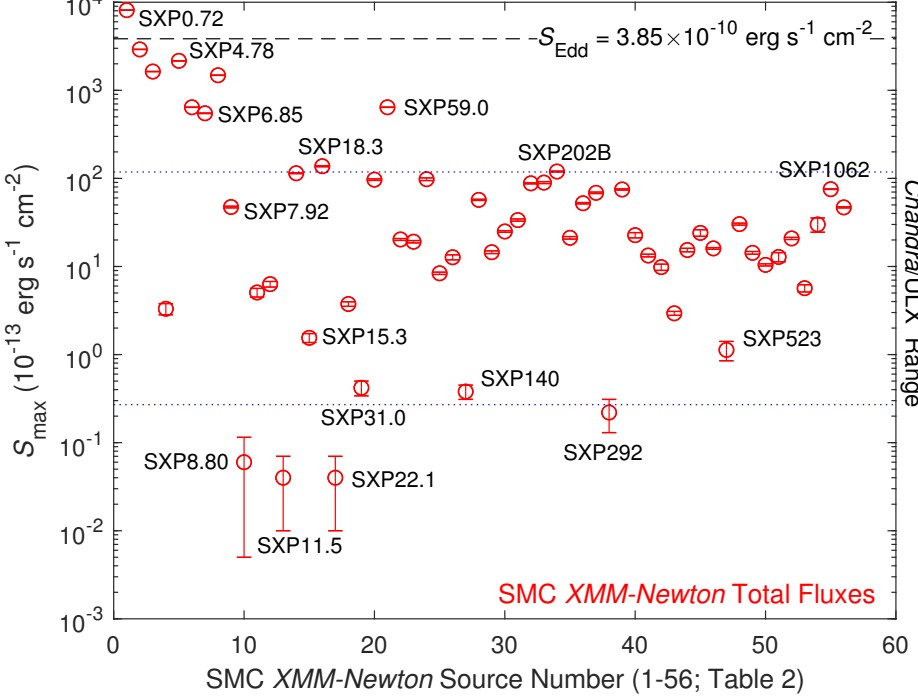

**Figure 5.** Sample of maximum X-ray fluxes for the SMC *XMM-Newton* sources in Table 2 (all 3 EPIC cameras combined). Errors are small in all but the 3 faintest sources. The range of ULX data and SMC *Chandra* data is virtually the same, and it is marked by the dotted lines. The Eddington flux $S_{\mathrm{Edd}}$ for the SMC is also shown as a dashed line. Here $S_{\mathrm{Edd}} = L_{\mathrm{Edd}}/(4\pi d^2) = 3.85 \times 10^{-10}$ erg s$^{-1}$ cm$^{-2}$, where we used $L_{\mathrm{Edd}} = 1.77 \times 10^{38}$ erg s$^{-1}$ for a canonical neutron star with mass $1.4 M_\odot$ and a distance of $d = 62$ kpc to the SMC [41].

Figure 5 also shows superimposed the actual range of the ULX data and the SMC *Chandra* data (as dotted lines) which, surprisingly, is very much the same for these two smaller samples. It is obvious that the X-ray flux values $S_{max}$ of most sources in all main samples (1–3) lie between $2.7 \times 10^{-14}$ erg s$^{-1}$ cm$^{-2}$ (log-value $-13.57$) and $1.2 \times 10^{-11}$ erg s$^{-1}$ cm$^{-2}$ (log-value $-10.92$). This defines a range of commonly measured X-ray fluxes for all of our samples that has not been previously highlighted for strongly variable X-ray point sources of any type. We think that this is a remarkable result. Figure 5 clearly shows that maximum ULX fluxes are nothing to brag about; several SMC HMXBs (SXP 0.72, 4.78, 6.85, 59.0, to name a few) rise to much higher intrinsic X-ray fluxes during their outbursts. From the point of view of X-ray fluxes, ULX sources appear to be modest, which indicates that they may be quite average HMXB sources whose apparently extreme X-ray luminosities are solely due to their enormous cosmic distances.

For example, the strongest by far ULX flux output (log-value $\approx -11$) comes from IC342 X-2 (Figure 3) at a distance of merely 3 Mpc, whereas the strongest X-ray luminosity ($L_{X,max} = 6217\, L_{Edd}$) comes from a modest source (IC3212) with a maximum log-value flux of only $\approx -12$, simply because this source happens to lie at the enormous distance of 101 Mpc. Here, for the Eddington luminosity, we use a value of $L_{Edd} = 1.77 \times 10^{38}$ erg s$^{-1}$ for a canonical neutron star with mass $1.4 M_{\odot}$. Another striking example is NGC891 ULX1 (Figure 3) that shows a large flux (log-value $-11.4$), but its distance is merely 9 Mpc, leading to an isotropic X-ray luminosity of only $L_{X,max} = 225\, L_{Edd}$, about 28 times smaller than that of IC3212. Based on these Song et al. results [1], strongly variable ULX sources appear to be overvalued in our current thinking.

## 3. X-Ray Flux/Luminosity Functions

In this section, we undertake the task of least-squares fitting of the data sets in order to compare their X-ray flux/luminosity functions.

### 3.1. ULX Sources

Figure 6 shows the number of observed ULX flux values $N(> S_{max})$ above a particular flux level of $S_{max}$. The data are consisent with a best-fit line with a slope close to $-1$. To be precise, the best-fit slope is determined to be $-0.9103 \pm 0.0574(1\sigma)$ (correlation coefficient $r = -0.9807$), with a $p$-value statistic of 0.0401 (No. 1 in Table 3), better than the $2\sigma$ confidence level. Such a steep slope has never been observed in a clean HMXB data set. It is typical of the slopes found for the disk populations of nearby galaxies ($\approx -1$) such as in N300, M31, and N1332 (Reference [26] and references therein). These populations are contaminated by background AGN and LMXBs which have been filtered out from our pure-HMXB SMC samples (see HMXB catalogs in References [42,43]). Knowing that the SMC data contain only HMXBs offers a clean "baseline" sample and a huge advantage in comparisons with other extragalactic X-ray samples. The difference in slopes is apparently fundamental; it was also found by Kilgard et al. [32] between samples of X-ray point sources from three young starburst galaxies and X-ray samples from four nonstarburst spiral galaxies, the latter of which show consistently steeper slopes of $\approx -1$.

The errors in Figure 6 (the grey areas) and subsequent similar diagrams are calculated according to the prescription of Gehrels [44]. The high end of the error bar in $N$ inside each bin is obtained from the equation [30]

$$N_{high} = N + \left(1 + \sqrt{N + 0.75}\right),$$
(1)

and the low end is obtained from the equation

$$N_{low} = N - \left(\sqrt{N - 0.25}\right) \qquad (N \geq 1).$$
(2)

We found a comparable result by modeling the maximum X-ray luminosities $L_{X,max}$ of the ULX sources listed in Reference [1]. We determined a best-fit slope of $-0.8404 \pm 0.0581(1\sigma)$ (correlation

coefficient $r = -0.9747$), with a $p$-value statistic of 0.0439. The small difference in slopes ($\sim 8\%$) between $-0.91$ and $-0.84$ is an indication of how much the errors in the distances $d$ affect the calculated isotropic $L_{X,max}$ values ($L_{X,max} \propto d^2$). These distance-related errors are certainly not present in our $S_{max}$ flux data sets.

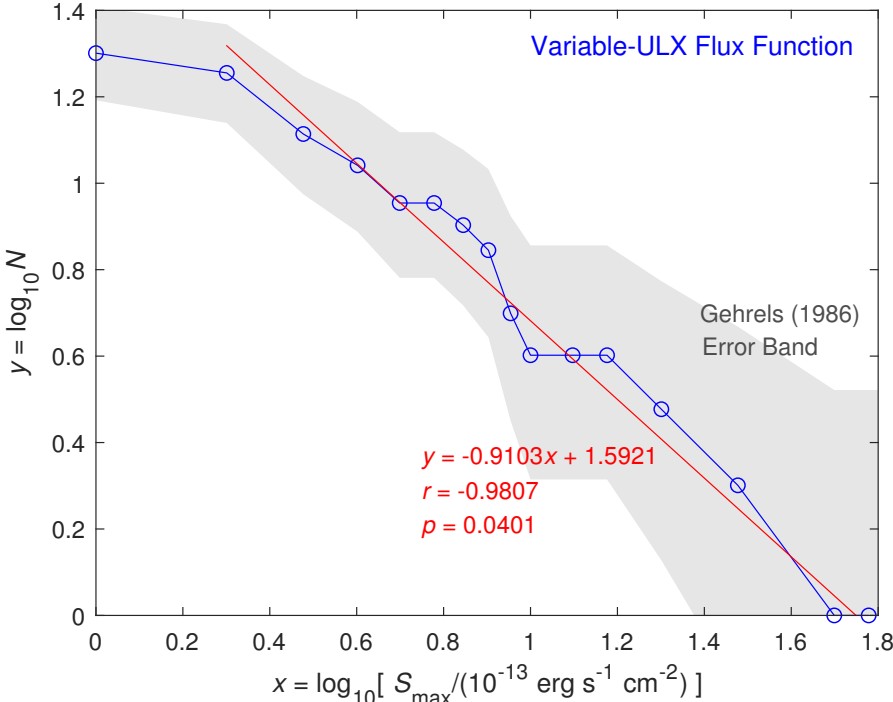

**Figure 6.** Cumulative number $N$ of ULX sources above a flux level $S_{max}$ versus $S_{max}$ on logarithmic scales (blue points). The data indicate a strong linear correlation (correlation coefficient $r = -0.9807$) with a steep slope of $-0.9103$ and a $p$-value statistic of $p = 0.0401$, better than the $2\sigma$ confidence level (red line). The grey area respresents the $1\sigma$ error bars to the data points according to the prescription of [44]. The errors in the best-fit line are listed in Table 3.

Based on previous results (cited in Section 1) and current HMXB results (sample No. 4-IR in Table 3), we conclude that the flux/luminosity function of variable ULX sources with slope $-0.91$ is different than that of pure SMC XMHB sources and that the ULX sources do not appear to be the high end of known nearby HMXBs with slopes of $\approx -0.4$.

### 3.2. SMC Sources

Figure 7 shows the number of observed SMC *Chandra* flux values $N(> S_{max})$ above a particular flux level of $S_{max}$ (Table 1). There are two kinks or "breaks" in the power-law fits, one at log-value 0.95 ($S_1 = 8.9 \times 10^{-13}$ erg s$^{-1}$ cm$^{-2}$) and another at log-value 1.9 ($S_2 = 7.9 \times 10^{-12}$ erg s$^{-1}$ cm$^{-2}$), albeit with fewer available points. The lower kink $S_1$ defines our completeness limit (see also Reference [26]). The higher kink $S_2$ (the broken power-law) is a feature usually observed in these types of diagrams (e.g., N300; [29,30]), but it is derived from few data points and its statistical significance is uncertain. The slope at intermediate values ($-0.6338$) is consistent with the average value ($\approx -0.6$) obtained for X-ray source samples that are not cleaned to eliminate LMXBs and/or background AGN (see discussion in Section 1 and Reference [25]). Our *Chandra* sample (Table 1) is clean, but it contains a small number of outbursting sources (41). This is because *Chandra* has never surveyed the Magellanic Clouds repeatedly, unlike the multi-year campaign undertaken by the *XMM-Newton* telescope.

Figure 8 shows the number of observed SMC *XMM-Newton* flux values $N(>S_{max})$ above a particular flux level of $S_{max}$ (Table 2). The same kinks appear in this figure at the same values as in Figure 7 (log-values of 0.95 and 1.9). However, the slope at intermediate values ($-0.4469$) is

different, and it seems to be more consistent with the lower value in SMC sources (−0.37) determined by [25] when they modeled only the securely identified HMXBs in the SMC. In either case, these results are very much different from those for ULX sources. Once again, the results strongly support the hypothesis that ULX sources are not the high end of HMXBs [11].

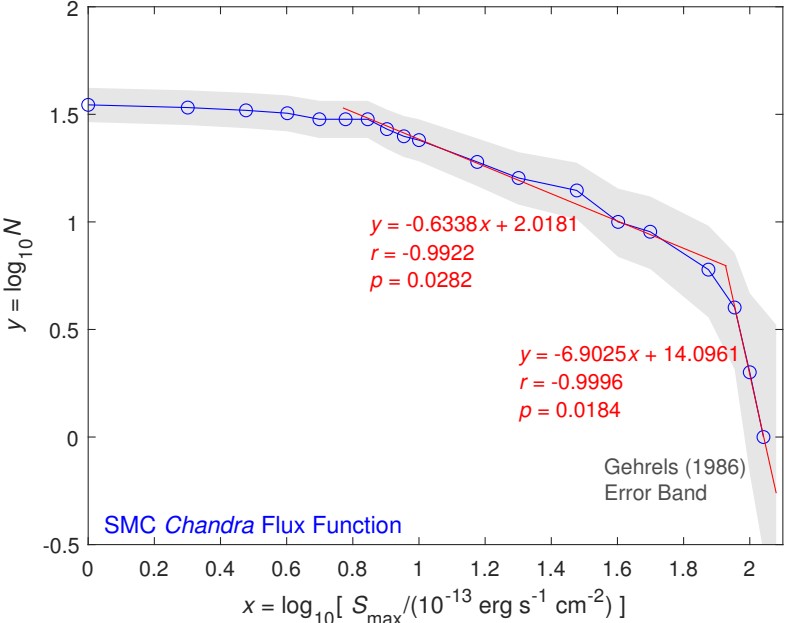

**Figure 7.** Cumulative number $N$ of SMC *Chandra* sources above a flux level $S_{max}$ versus $S_{max}$ on logarithmic scales (blue points). The grey area respresents the $1\sigma$ error bars to the data points according to the prescription of [44]. The errors in the best-fit line are listed in Table 3.

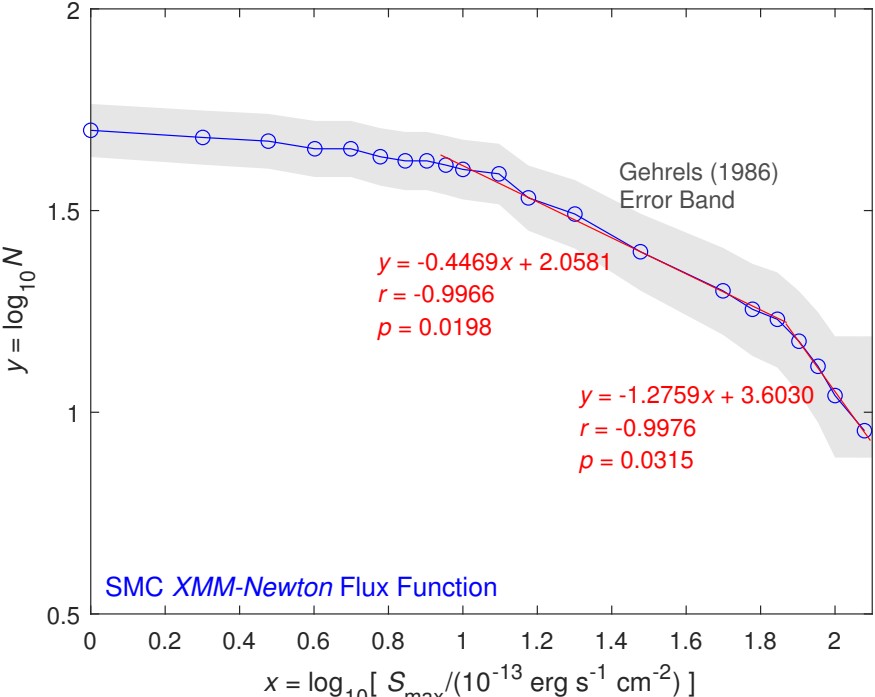

**Figure 8.** Cumulative number $N$ of SMC *XMM-Newton* sources above a flux level $S_{max}$ versus $S_{max}$ on logarithmic scales (blue points). The grey area respresents the $1\sigma$ error bars to the data points according to the prescription of [44]. The errors in the best-fit line are listed in Table 3.

When we combine the two SMC data sets (2 + 3), the new sample 4 contains 58 unique SMC maximum X-ray fluxes (36 from *XMM-Newton* data and 22 from *Chandra* data; a combination of Tables 1 and 2) and the best-fit slope at intermediate flux values turns out to be $-0.3939 \pm 0.0195\,(1\sigma)$, in better agreement with the findings of Reference [25] for pure HMXB sources. In this least-squares fit, we also find that $r = -0.9915$ and that $p = 0.0315$ (Table 3), so our conclusions appear to be statistically solid for the combined SMC HMXB sample 4, and they are in good agreement with previous results from the SMC.

## 4. Two-Sample Kolmogorov–Smirnov Tests

Here we present the results from two-sample Kolmogorov–Smirnov (KS) tests that we performed in our data sets. The two-sample KS test compares two samples and examines the null hypothesis ($H_0$) that they are both derived from the same continuous parent distribution. The alternative hypothesis ($H_1$) is that the samples are not derived from the same continuous distribution, with no knowledge of what such parent distributions might be. The two-sample KS tests are valid for our paired samples with sizes $n_1$ and $n_2$ since all of our $(n_1, n_2)$ pairs satisfy the condition that

$$\bar{n} = \frac{n_1\, n_2}{n_1 + n_2} \geq 4\,, \tag{3}$$

by large margins (in our samples, $\bar{n}_{\min} = 15.53$).

We compared the ULX sample 1 versus the two main SMC samples (2 and 3), and then we also compared the main SMC samples against one another. The results are as follows (top part of Table 4):

1. The ULX data set is not derived from the same continuous distribution as any one of the SMC data sets at a confidence level of $\alpha = 0.05$. We reject the null hypothesis $H_0$ at probability levels of $p = 4.7 \times 10^{-3} - 1.4 \times 10^{-6} << \alpha$. The $D$ statistic values (the largest deviation in cumulative probabilities between the two samples) are also consistently larger than the critical values $D_{\mathrm{crit}}$ of the KS tests (Table 4), which also leads to rejection of $H_0$. Here, we calculate the critical values $D_{\mathrm{crit}}$ of the $D$ statistic for $\alpha = 0.05$ [45] from the equation

$$D_{\mathrm{crit}} = c(\alpha)\sqrt{1/n_1 + 1/n_2} = 1.35810\sqrt{1/n_1 + 1/n_2}\,, \tag{4}$$

   where $n_1$ and $n_2$ are the sizes of the two paired data sets. The coefficient $c(\alpha = 0.05) = 1.35810$ is determined from the the inverse of Equation (15) given by [45] in their Section 3.3.1, viz.

$$c(\alpha) = \sqrt{-0.5 \ln(\alpha/2)}\,. \tag{5}$$

   If $D < D_{\mathrm{crit}}$, then we accept the null hypothesis $H_0$, but this not the case here. The null hypothesis is clearly rejected since $D > D_{\mathrm{crit}}$ for all ULX cases listed at the top section of Table 4.

2. The two main SMC data sets (2 and 3) are derived from the same continuous distribution. This result makes sense since *XMM-Newton* and *Chandra* have been looking at the same exclusive group of SMC HMXB sources for more than 20 years, albeit at different campaigns and exposure times. The asymptotic $p$-value of the two-sample $D$ statistic is $p = 0.308 > \alpha$, and the MATLAB $D$ statistic agrees since $D = 0.193 < D_{\mathrm{crit}} = 0.279$ (Table 4). Figure 2 also shows that the two distributions are quite similar. Thus, the null hypothesis $H_0$ is accepted for the two main SMC data sets at the $\alpha = 0.05$ confidence limit. Indeed, they are derived from the same continuous distribution (although this is not a normal distribution; see bottom part of Table 4).

We have constructed the cumulative distribution functions (CDFs) for the two main SMC data sets 2 and 3, and we measured a $D$ statistic of 0.184 (Figure 9). This value is slightly smaller than that produced internally by the MATLAB `kstest2` routine (0.193), which does not output the CDFs. The difference in $D$ statistic values ($< 5\%$) is probably due to the chosen bin sizes; it does not appear to

be significant, and our decision to accept the null hypothesis at the $\alpha = 0.05$ confidence level appears to be solid.

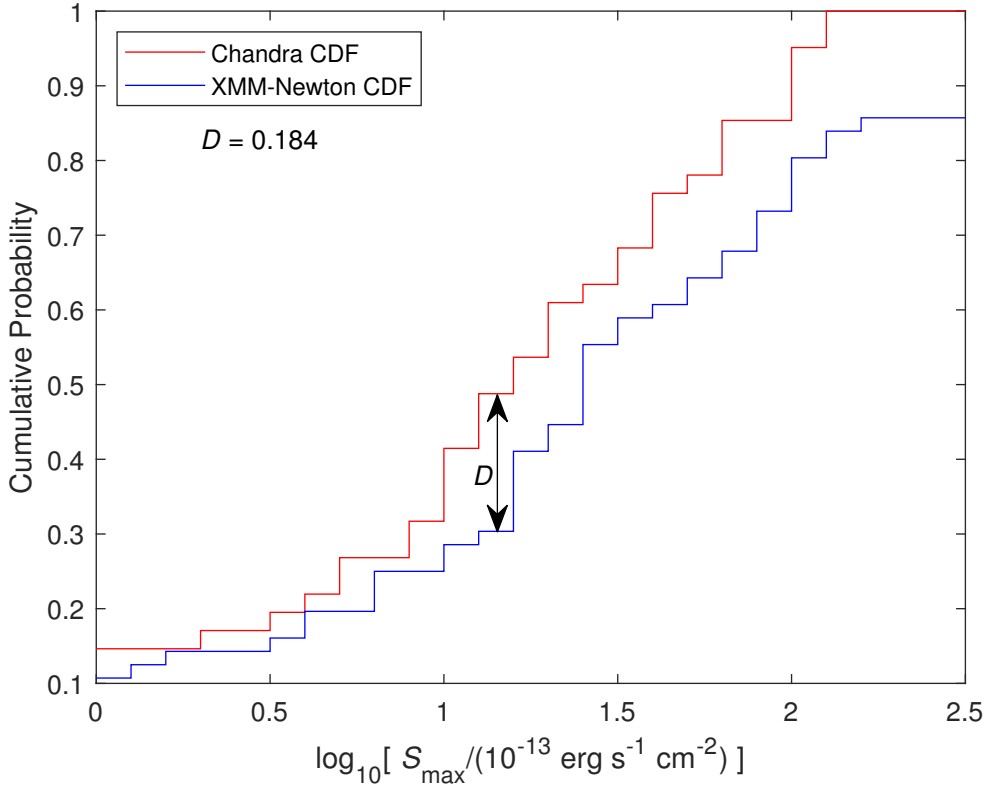

**Figure 9.** Cumulative distribution functions for the SMC data sets (2 and 3). The KS $D$ statistic is determined by the largest deviation between the two cumulative distributions. We find that $D = 0.184$, slightly lower than the value determined by the internal MATLAB routine `kstest2` (0.193). Still, the null hypothesis $H_0$ must be accepted since this $D = 0.184 < D_{\mathrm{crit}} = 0.279$.

## 5. A Timid Look into X-ray Luminosities and ULX Cosmic Distances

Figure 10 shows the maximum X-ray luminosities for our 3 main samples (1–3) introduced in Section 2. The errors in the ULX data are not known, but they are certainly dominated by errors in distances $d$ since errors in fluxes are extremely small (Figures 3–5). Although the ULX distance errors are generally much larger, there also exist smaller intrinsic errors of up to $\pm 10$ kpc in the distances of HMXB sources within the SMC [46]. This is why we mostly analyzed X-ray fluxes in this work, where we avoided such distance-related errors up to this point.

Figure 11 shows the maximum X-ray fluxes of the ULX sources versus cosmic distance $d$ in Mpc. Most of these sources have fluxes below the modest level of $S_{\max} = 2 \times 10^{-12}$ erg s$^{-1}$ cm$^{-2}$ (log-value 20). These sources do not seem to be impressive by any account (except by their extreme distance-related X-ray luminosities shown in Figure 10). In fact, we know from Figures 1 and 5 that their fluxes appear to be quite average compared to the *XMM-Newton* fluxes of HMXBs in the SMC. We reiterate that IC3212 shows in Figure 11 a modest flux (ten times smaller than the highest flux observed from IC342 X-2), but IC3212 turns out to be the most luminous source in the samples shown in Figure 10 because of its enormous distance of 101 Mpc. This example highlights the risk of relying on luminosities to interpret X-ray data, rather than looking at intrinsic properties of the sources such as flux measurements which are characterized by insignificant errors of all types (see Figures 3–5).

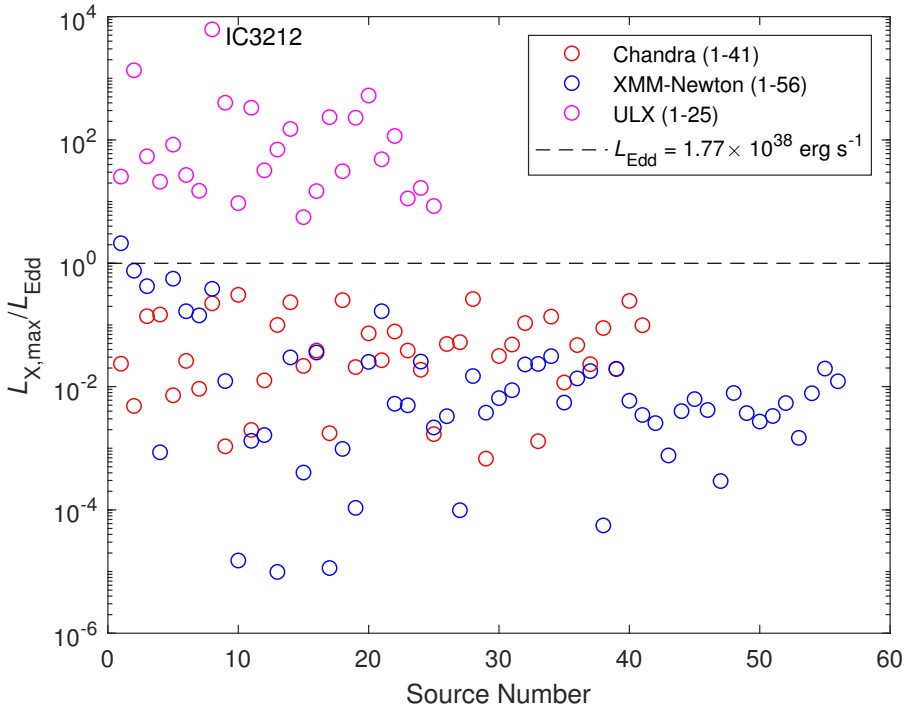

**Figure 10.** Maximum X-ray luminosities, scaled to the canonical Eddington luminosity $L_{Edd}$ of a $1.4 M_\odot$ neutron star, for our 3 main samples (1–3) in Section 2.

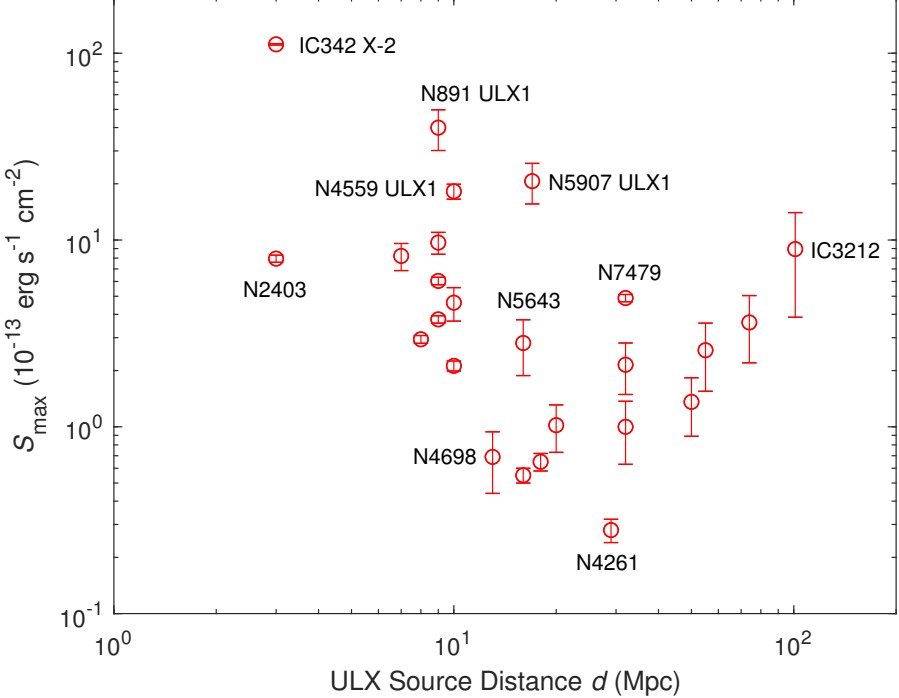

**Figure 11.** Maximum X-ray fluxes of the ULX sources from ULX data set 1 versus distance $d$ in Mpc [1]. The majority of these sources radiate below a flux level of $S_{max} = 2 \times 10^{-12}$ erg s$^{-1}$ cm$^{-2}$, which is nothing to brag about, despite their enormous distances that reach as far out as 101 Mpc.

Next we pretend to ignore our main result that the ULX and SMC samples are not derived from the same continuous distribution, and we combine the X-ray luminosities of data sets 1 and 4 into

a new $L_{X,max}$ pseudo-data set (set 5 in Section 2 and in Table 3). In data set 5, we effectively allow the ULX set 1 to "contaminate" the combined SMC set 4, or vice versa. The X-ray luminosities in all samples are calculated from the well-known equation for isotropic emission

$$L_{X,max} = \left(4\pi d^2\right) S_{max} . \tag{6}$$

Figure 12 shows the number of observed X-ray luminosity values $N(> L_{X,max})$ above a particular level of $L_{X,max}$ versus the ratio $L_{X,max}/L_{Edd}$ on logarithmic scales (blue points). The statistical results for this data set and for its X-ray luminosity function are listed at the bottom rows of Tables 3 and in Table 4. We find that:

1.  The one-sample KS test shows that sample 5 is not drawn from a normal distribution (Table 4).
2.  The completeness limit of the sample (Figure 12) is located at a log-value of 0.954 ($L_{X,max} = 9.0\ L_{Edd} = 1.6 \times 10^{39}$ erg s$^{-1}$). This value is comparable to the critical value that empirically separates HMXBs from ULX sources [11,21].
3.  The second kink observed at higher values in the SMC samples is gone. So we can fit the X-ray luminosity function of data set 5 with a single power law of slope $m$ beyond the completeness limit.
4.  The slope $m$ in the luminosity function (Figure 12) lies between the slopes of data sets 1 and 4 (Table 3); we find that $m = -0.5716 \pm 0.0240(1\sigma)$, which is close to the average value ($-0.6$) obtained from "contaminated" HMXB samples that contain also other types of X-ray sources (see Sections 1 and 3.2).

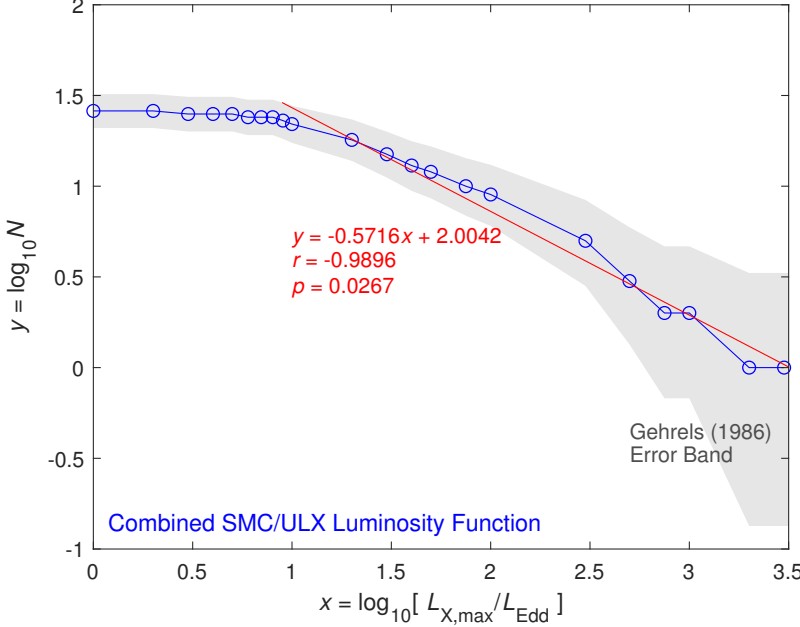

**Figure 12.** Cumulative number $N$ of sources in the combined ($1 + 4 = 5$) data set above a luminosity level $L_{X,max}$ versus $L_{X,max}/L_{Edd}$ on logarithmic scales (blue points). The grey area respresents the $1\sigma$ error bars to the data points according to the prescription of [44]. The errors in the best-fit line are listed in row 5 of Table 3.

In another experiment, we paired up pseudo-sample 5 with sample 1 and then with sample 4. We performed two-sample KS tests in order to find out whether the paired data sets (5-1 and 5-4) could originate from the same continuous parent distribution (null hypothesis $H_0$). At the $\alpha = 0.05$ confidence level, the results (not listed in Table 4) point to a clear rejection of $H_0$. The asymptotic $p$-values of the $D$ statistic are much smaller than $\alpha$ in both cases. We found that $p = 4.0 \times 10^{-9}$ for

the 5-1 samples and that $p = 3.0 \times 10^{-3}$ for the 5-4 samples. The latter somewhat high $p$-value is understood because sample 5 is dominated by the SMC sources of sample 4, and sample 4 is that of the combined SMC sources—so the contamination of sample 5 by ULX sources is minimal. Yet, the two-sample KS test finds that sample 5 is sufficiently contaminated to not be related statistically to the pure SMC/HMXB sample 4.

## 6. Discussion

The statistical results support our main conclusions that ULX sources and SMC Be/X-ray sources originate from different continuous parent distributions and that these are not normal or log-normal distributions. We find that pure HMXB sources show a flux/luminosity broken power-law function of the form $N \propto (S_{\max})^m$ with dual slopes of $m \approx -0.4$ and $m \approx -2.0$ (the latter is uncertain due to using few data points), as in the No. 4 rows of Table 3. The break in the second power law occurs at a log-value of 1.9 ($S_2 = 7.9 \times 10^{-12}$ erg s$^{-1}$ cm$^{-2}$) (see also Figures 7 and 8). On the other hand, ULX sources show only a single flux/luminosity power-law function with slope $m \approx -0.91$ (Figure 6) comparable to slopes of $m \approx -1$ seen in nonstarburst galaxies and for the disk populations of nearby spiral galaxies (see [26,29,32] and references therein). This may not be just a coincidence. We took the following steps to investigate the apparent agreement between those slopes around the value of $m = -1$:

(a) First, we considered the Carpano et al. [26] X-ray fluxes of the point sources in N300 within the $D_{25}$ isophote of the optical disk of the galaxy (60 point sources with counts above 20) for which the slope of the flux/luminosity function is $m = -1.17 \pm 0.17$ (comparable to that of our ULX sample),[1] and we ran a two-sample KS test against the Song et al. [1] variable ULX sources. The KS test clearly shows that the the two data sets are not derived from the same continuous distribution, and one-sample KS tests show that the N300 data are not derived from a normal or a log-normal distribution. The rejection of the null hypothesis $H_0$ occurs in the two-sample KS test at a particularly strong level of an asymptotic $p$-value of $p = 3.1 \times 10^{-15} \ll \alpha = 0.05$, and the KS $D$ statistic is $D = 0.950 \gg D_{\mathrm{crit}} = 0.323$ (Equation (4)). This result indicates that the Song et al. [1] sample of variable ULX sources is not related to the X-ray samples derived from the optical disks of nearby spiral galaxies such as the disk of N300, despite the comparable slopes of the luminosity functions.

The fluxes of X-ray point sources in N300 shown in both References [26,29] are very low compared to the Song et al. [1] ULX flux values. There is no significant overlap between the N300 samples and the ULX sample, which makes the results of the two-sample KS statistic totally understandable: The CDFs of the N300 samples approach a value of order 1 long before the ULX CDF even begins to rise significantly above the starting value of zero (and then $D \lesssim 1$). Other nonstarburst spiral galaxies [32,33] may very well have similar populations of X-ray point sources as N300, but the ULX sources are not randomly chosen from such a collection of galaxies. The Song et al. ULX sample is formed from the sources with the highest fluxes ever observed in each particular galaxy; therefore, the ULX data only sample the very high end of the X-ray point-source population in each particular galaxy. Then, the ULX sample is a sample of the highest flux values in nonstarburst galaxies, although these fluxes are barely comparable to the average HMXB fluxes observed in the SMC starburst (as in the ULX flux range shown in Figure 5 by dotted lines).

(b) Second, we constructed 50 simulated data sets each with "fluxes $F(i)$" distributed uniformly between $i = 1$ and various $i = n - 1$ maximum values, where $n$ is even and $14 \leq n \leq 112$ (each

---

[1] A detailed *Chandra* study of N300 by Binder et al. [29] resulted in comparable results: at the 0.5–2 keV band, the statistical X-ray luminosity function had an overall slope of $-1.03 \pm 0.10$ and that of the pure-HMXB subsample had a slope of $-0.86 \pm 0.19$.

sample size was set to $n/2$). The slope of the $F(i)$ relation in each set was preset to $m = -1$ in the generating function

$$F(i) = n + m \times i. \tag{7}$$

These 50 data sets represent populations of X-ray sources in various spiral galaxies. Next we created a new data set with 50 values (representing the ULX sample) for which we picked the maximum value $F_{max}$ from each "galaxy", and we calculated the slope of the resulting $\log N$-$\log F_{max}$ distribution at intermediate (30–80) flux values. An example from a simulation is shown in Figure 13, where the determined slope is $m \approx -0.95$.

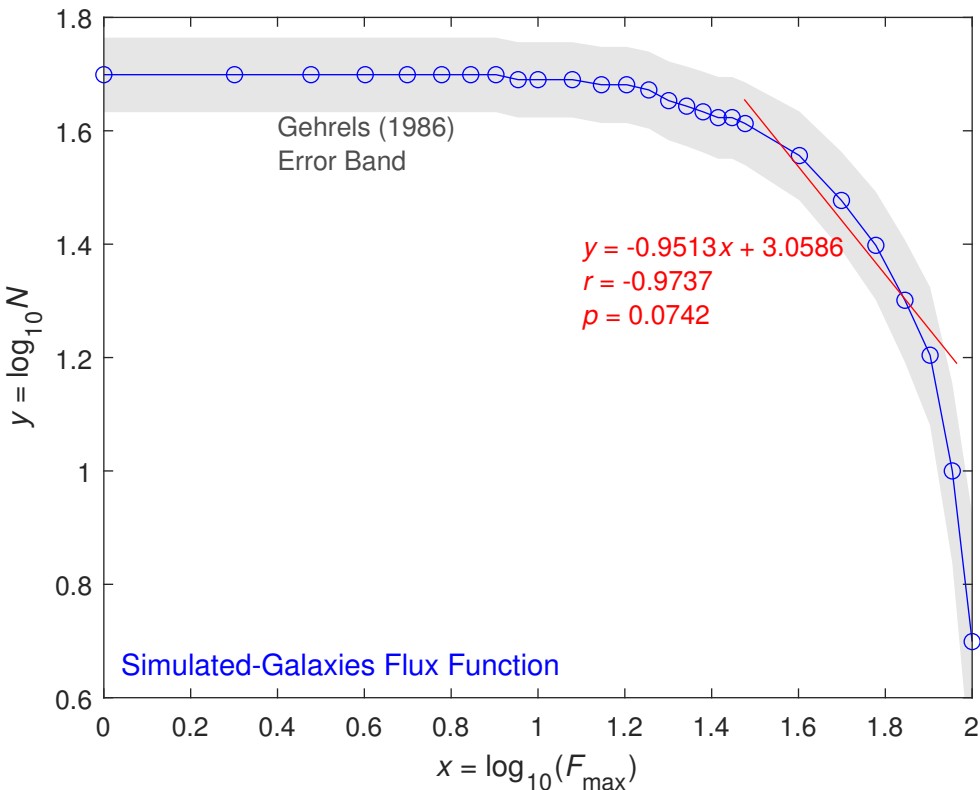

**Figure 13.** A typical $\log N$-$\log F_{max}$ diagram derived from one simulation of 50 "galaxy" samples. The measured slope at intermediate $F_{max}$ values (30–80) is $m = -0.9513 \pm 0.1113(1\sigma)$.

We ran 10,000 such simulations several times over, and each time we analyzed statistically the resulting 10,000 slopes. The distribution of slopes is similar among these repeated runs. A typical outcome is shown in Figure 14. There is a strong preference for 40% of the values to aggregate near $m = -1$. There are also secondary peaks near $m = -1.2$ and $m = -0.8$. The one-sample KS test indicates that these peaks are sufficient to make the sample not be related to the normal distribution. The majority of slopes (63%) are concentrated in the range of $m = -1 \pm 0.2$, and about 90% of the slopes are found within the range of $m = -1 \pm 0.25$.

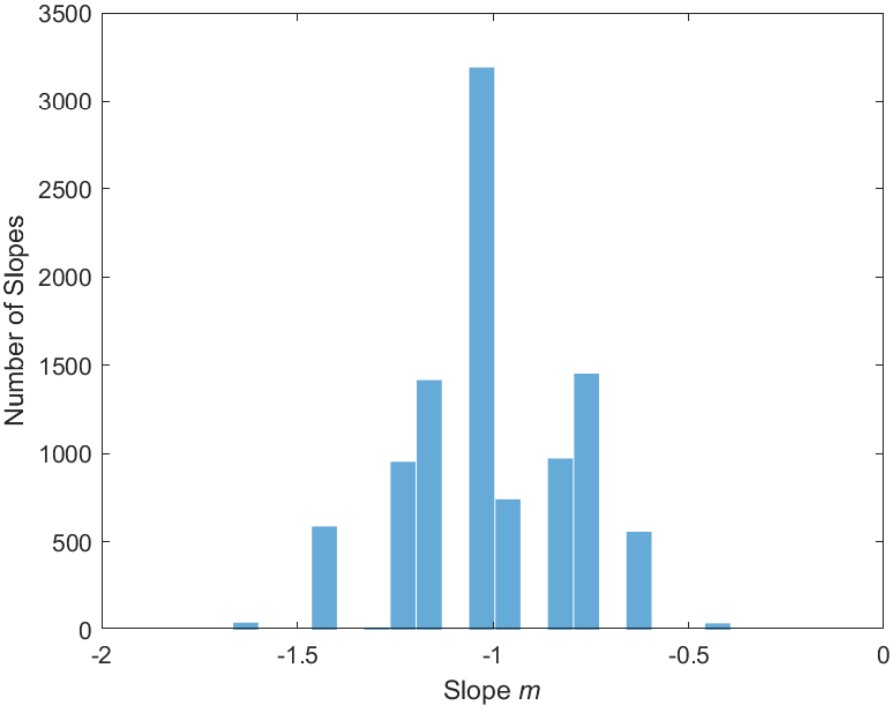

**Figure 14.** Histogram of 10,000 slopes $m$ measured from $\log N$-$\log F_{max}$ diagrams in 10,000 typical simulations of 50 "galaxy" samples each. About 40% of the slopes aggregate at $m = -1$, and the majority of slopes (90%) fall to within $\pm 0.25$ of $m = -1$.

We also ran additional experiments in which the above trends did not materialize as clearly:

(1)    When we changed the preset slope of the 50 initial samples of "galaxies" to $-0.5$, we obtained $\log N$-$\log F_{max}$ diagrams with slopes aggregating near two or three unrelated values.

(2)    When we chose the preset slope of the "galaxy" samples randomly between $-0.85$ and $-1.2$, some experiments produced $\log N$-$\log F_{max}$ slopes peaking near $m = -1$ (as in Figure 14), but others did not show this trend.

(3)    When we expanded the preset range of slopes to $(-1.6, -0.4)$, aggregation of slopes at $m = -1$ did not occur. An example of this case is shown in Figure 15, where the 10,000 $\log N$-$\log F_{max}$ slopes are distributed about equally within a range of $m$-values.

(4)    When we repeated the simulations with a random y-intercept in the generating Function (7) ($\max(i)$ plus a positive random number $\leq 3$), the qualitative properties of the above histograms did not change in a substantial manner, although the $\log N$-$\log F_{max}$ slopes spread out to nearly all the bins.

Thus, it seems that the apparent agreement between the ULX result ($m = -0.91$) and the results from nearby spiral galaxies and nonstarburst galaxies ($m \approx -1$) may not be coincidental.

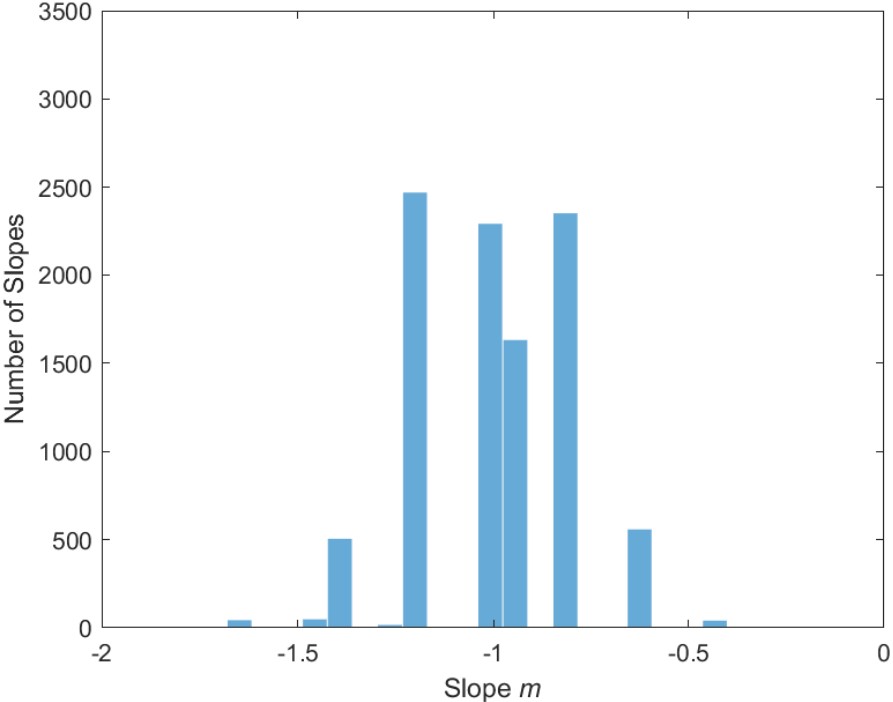

**Figure 15.** As in Figure 14, but the slopes of the 50 "galaxy" samples were chosen randomly from the interval $(-1.6, -0.4)$. The slopes $m$ of the 10,000 log $N$-log $F_{max}$ diagrams do not aggregate at $m = -1$, where we find only 23% of them (furthermore, 25% of the slopes are at $m = -1.2$ and 24% are at $m = -0.8$).

## 7. Summary

We revisited the stringently constrained but very valuable ULX data set of Song et al. [1] of strongly variable ULX sources. These are 25 0.3–10 keV X-ray sources that vary by more than a factor of 10 between their high and low emission states, so their variability is akin to that of SMC HMXB sources. We compared this variable ULX sample to the variable HMXB samples from our latest SMC library that produced 41 *Chandra* 0.3–8 keV sources and 56 *XMM-Newton* 0.2–12 keV sources. We worked mostly with X-ray fluxes because their errors are very small (Figures 3–5), and they are not affected by the large errors inherent to the distances to ULX sources.

The main advantage of using the SMC samples as a benchmark is that all of their sources are clearly identified as Be/X-ray binaries, and these samples are not contaminated by other types of X-ray point sources such as LMXBs and background AGN (see the compilations in References [42,43]). With this in mind, we confirmed the slopes of the luminosity functions ($-0.37$ to $-0.6$) previously found in various HMXB samples and in young starburst-galaxy samples as well as in the SMC (details are given in Table 3 and in Sections 1 and 3.2).

On the other hand, the slope in the fluxes of the ULX sample [1] is significantly steeper ($-0.91$) and more similar to those found for the disk populations of nearby spiral galaxies and in nonstarburst galaxies (Section 3.1). This slope implies that there is a marked deficit of variable ULX sources at higher X-ray flux values. This fact alone tells us that ULX sources are not exceptional, since they emit small or modest amounts of X-ray photons as compared to the brightest SMC sources. When we used the X-ray luminosities $L_{X,max}$ listed in Reference [1], the slope of the luminosity function changed to $-0.84$ (Section 3.1). This 8% difference in slopes is representative of the errors $\Delta d$ in the distances $d$ to the ULX sources since $L_{X,max} \propto d^2$ (Equation (6)) and the luminosity errors are $\Delta L_{X,max} \propto 2 \Delta d$, whereas the errors in fluxes and photon counts are negligible.

We carried out formal KS tests in order to compare our samples between one another and against the normal distributions (Section 4 and Table 4). The one-sample KS tests show that no sample comes from a normal distribution or from a log-normal distribution. The two-sample KS tests show that only the two SMC samples come from the same continuous parent distribution (that of the SMC). We conclude that the ULX sample is not related to the SMC HMXB samples. This is contrary to the expectation that ULX sources could be the high end of the HMXB distribution [11].

We also ran an experiment in which we contaminated the X-ray luminosities of our combined SMC sample 4, clean and full of HMXBs, by concatenating the X-ray luminosities of the ULX sample (Section 5). We carried out KS two-sample tests between this pseudo-sample and the SMC and ULX samples. The null hypothesis was rejected squarely at the $\alpha = 0.05$ confidence level in both cases. Apparently, the 25 sources of the ULX sample are sufficient to contaminate the CDF of the SMC sample 4 and render it "too different" (meaning that the SMC sample, the ULX sample, and the pseudo-sample all appear to come from different continuous parent distributions).

Finally, we investigated the agreement between slopes ($m \approx -1$) in the flux functions of X-ray sources in nearby spiral galaxies (Reference [26] and references therein) and in the Song et al. [1] ULX sample (Section 6). Simulations of 50 "galaxy" data sets with a preset slope of $m = -1$ indicate that the log $N$-log $F_{max}$ diagrams of the maximum "fluxes $F_{max}$," where each $F_{max}$ value is obtained from each simulated "galaxy" data set, show a preference to mimic the same slope of $m = -1$. In particular, 63% of the simulated slopes fall in the range of $m = -1 \pm 0.2$, and about 90% of the slopes lie within $\pm 0.25$ of $m = -1$ (Figure 14). Thus, it appears that our ULX result ($m = -0.91 \pm 0.057(1\sigma)$) is not a mere coincidence when compared to the X-ray samples obtained from nearby spiral galaxies such as N300 [26,29] and in nonstarburst glaxies [32]. There could be a link between these populations despite the global distribution of ULX sources and unrelated to their presumed HMXB nature (because [33] advocates for a universal HMXB slope of $m \approx -0.6$, albeit in starburst galaxies).

**Author Contributions:** All authors have contributed equally to this article. All authors have read and agreed to the published version of the manuscript.

**Funding:** This research was funded in part by NASA ADAP grants NNX14-AF77G and 80NSSC18-K0430.

**Acknowledgments:** The authors would like to thank the reviewers for their detailed comments and suggestions that led to significant improvements of the manuscript.

**Conflicts of Interest:** The authors declare no conflict of interest. The funders had no role in the design of the study; in the collection, analyses, or interpretation of data; in the writing of the manuscript, or in the decision to publish the results.

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
