# Peer review of "Variable Magellanic HMXB Sources versus Variable ULX Sources: Nothing to Brag about the ULX Sources"

_galaxies, doi:10.3390/galaxies8040070_

Round 1
Reviewer 1 Report
I have carefully read the manuscript
Variable Magellanic HMXB Sources versus Variable ULX Sources: Nothing to Brag about the ULX Sources
by Christodoulou et al.
The work is generally interesting but I think that some points, see below, have to be adjusted/clarified before it is accepted for publication. In particular, bothe the samle definition and the statistical analysis must be detailed in a more precise way.
I am willing to review a revised version of the manuscript with all the points below addressed
line 1: flux variabilities in the 0.3-10 keV X-ray energy range
l3: dependence on what? I would say the Log N- Log S relation
l5: define what "a young X-ray sample" ....
l6: clearly show
l6: "in our Universe" ... this is a too ample definition. what is the space distribution of the ULX sample considered in this work? As far as one can understand from the abstract, these ULXs are all in the SMC (see line 8)
l9: specify that it is the Log N - Log S distribution
l12: specify what alpha is
l13: explain why the two values of p are so different for the XMM and Chandra samples
l16: the ULX sample is in the SMC. how do starburst galaxies fit in the context?
l16: what are the two data sets?
l25: Song et al. distilled ....
l26 and l1: variability by a factor of .....
l29: why is this group of ULX "exotic" ?
table 1 and 2: spell SXP .... with which spectral model did you compute fluxes? what are the numbers in the SXP column?
l38: extragalactic X-ray sources
l48: "point sources" is too generic
l42: I would say "effects in the ULX emission and/or systematics in the ULX observations"
l43: I would rephrase as "are detected because they are beaming" ..this would be a clear selection effect
l45: clarify what "best-fit line" is
l49: Magellanic: SMC, LMC or both?
l50: of course, you have to model X-ray fluxes before modelling X-ray luminosities
l61: specify whether the ULX sources are only in the SMC or also in other galaxies
l63: do the chandra and xmm samples overlap?
l71: the XMM-Newton and Chandra samples
l72: Both of them ....
l74: Is the presence of secondary peaks due to an observational bias?
l80: Flux variability ....
Fig. 4 and 5 : what's the difference between Chandra fluxes and XMM-Newton total fluxes?
l91: ..clearly shows ...
l93: this sentence is unclear. if ULXs have high luminosities it means that they are not average sources
l96: what is log value?
l115-116: clarify this sentence
l117: sample of what in the non/starburst galaxies?
l142: different from those ....
page 14, bullet 1: define the D statistic values
l208: define what "parent distribution" you mean
l213: add their slopes
l253: 0.3-10 keV X-ray sources
l266: ... Universe --> in the considered galaxy sample
l267: specify in which sense ULXs are not special
Reviewer 2 Report
The main purpose of the work is not clearly stated and appears to be twofold:
1) Check the uniformity and isotropy of the distribution of a sample of highly variable ULXs in the Universe (so more related to cosmology).
2) Compare the logN-logS diagram of the ULX sample with the one of HMXBs in the SMC (so more related to X-ray binary population studies).
The methods used in part 2) is very doubtful (see comments later).
One important piece of information that is missing at the beginning of the paper, is that the ULX sample from Song2020 are pulsar candidates (whose pulsations could be found with more sensitive instruments), and that the HMXBs in the SMC are also all pulsars (none of them being ULX). Note that the currently confirmed ULX pulsars are all HMXBs. So although different accretion mechanisms are present, the nature of the sources in the two samples, is possibly to be the same (pulsars with massive companions).
A large part of the work is spent in comparing the flux distribution of the SMC HMXBs observed with Chandra on one hand and observed by XMM on the other hand, and conclude that they are consistent (see Figure 2). But isn't that
what one would expect from the beginning? The SMC XMM and Chandra sample should be merged from the beginning of the work (taking the highest flux from the two sample for each source, and eventually correcting for the small difference in energy bands). One paragraph at the beginning, explaining that the flux distribution of the sources observed with Chandra and XMM are consistent, is enough. The merging of the data is also useful because many sources have been observed only once or few times, making the value of the "maximum" source flux more uncertain.
The comparison of the logN-logS diagram of the ULX sample, with the one SMC one is doubtful as the authors do not correct for the large difference in distances (they should comparing luminosities instead of fluxes) and
for the difference in the star formation rate of the host galaxies (since a higher SFR would lead to a larger number of brighter X-ray sources, see Figure 3 in Grimm2003). The plots presented in the paper of Grimm2003, mentioned in the references, are much more meaningful and very relevant for this work. SFR of the host galaxies could probably be found in some online catalogue.
Only with these 2 corrections, one could compare properly the logN-logS diagram of the two samples and see if one is the continuation of the other, or if they overlay (Fig.1 is therefore also not very meaningful).
Other comments:
- The work on NGC 300 mentioned in Carpano 2003 is not really relevant for this work, because the X-ray population is a mixture of sources of different natures (SNRs, LMXBs, HMXS, background AGNs, etc).
Instead one could refer to the work of Binder et al. 2012 (ApJ,758,15) who made a more careful analysis of the NGC 300 X-ray population, and plotted the logN-logS diagrams for the different types of sources.
-The simulations performed in section 6, are also quite hard to understand, so I would suggest the authors to provide more details about the intermediate steps. For example can the authors show what a set of 50 simulated logN-logS curves look like? Do they have the same y-intercept (which would mean the authors assume the galaxies are all at the same distance, with a similar SFR)? Can the authors also show what the a typical logN-logFmax distribution created from the Fmax value from each galaxy look like? Also the preset value of -1 should be better justified as is depend on the population type studied (here presumedly mainly HMXBs).
In conclusion, the work as it is presented here cannot be accepted for publication as a substantial amount of corrections, as suggested here above, are necessary.
Round 2
Reviewer 1 Report
Dear Authors,
I am satisfied with the revision of the manuscript and with the work done to address all points raised in my previous report.
I am glad to recommend the manuscript for publication
Reviewer 2 Report
Dear authors,
I appreciate that some of my comments have been taken into account, as explicitly mentioning that sources from the ULX sample are expected to be HMXBs pulsars, and putting more details about what has been done in the simulations. I also agree that the Chandra and XMM SMC samples could be analysed separately, although I would have preferred to see the 2 samples merged.
However, I still do not agree with the authors on the method used, which compares flux functions of the SMC HMXBs and ULX HMXBs rather than luminosity functions, which would take into account the large differences in distance. I can therefore not accept this paper for publication, since in my opinion conclusions are meaningless.
I would therefore prefer to let the editor and/or other possible reviewers the decision about the paper acceptance.